# Systematic Review and Meta-Analysis of the Financial Impact of 30-Day Readmissions for Selected Medical Conditions: A Focus on Hospital Quality Performance

**DOI:** 10.3390/healthcare12070750

**Published:** 2024-03-29

**Authors:** Iwimbong Kum Ghabowen, Josue Patien Epane, Jay J. Shen, Xan Goodman, Zo Ramamonjiarivelo, Ferhat Devrim Zengul

**Affiliations:** 1Department of Healthcare Administration, School of Public Health, University of Nevada Las Vegas, Las Vegas Nevada, NV 89154, USA; iwimbong@unlv.nevada.edu (I.K.G.); jay.shen@unlv.edu (J.J.S.); 2Department of Healthcare Administration, School of Public Health, Loma Linda University, Loma Linda, CA 92354, USA; jepane@llu.edu; 3University Libraries, School of Public Health, University of Nevada Las Vegas, Las Vegas Nevada, NV 89154, USA; xan.goodman@unlv.edu; 4School of Health Administration, College of Health Professions, Texas State University, San Marcos, TX 78666, USA; zhr3@txstate.edu; 5Department of Health Services Administration, School of Health Professions, University of Alabama Birmingham, Birmingham, AL 35294, USA

**Keywords:** 30-day readmission, hospitals, cost, meta-analysis, quality improvement, financial performance

## Abstract

Background: The Patient Protection and Affordable Care Act (ACA) established the Hospital Quality Initiative in 2010 to enhance patient safety, reduce hospital readmissions, improve quality, and minimize healthcare costs. In response, this study aims to systematically review the literature and conduct a meta-analysis to estimate the average cost of procedure-specific 30-day risk-standardized unplanned readmissions for Acute Myocardial Infarction (AMI), Heart Failure (HF), Pneumonia, Coronary Artery Bypass Graft (CABG), and Total Hip Arthroplasty and/or Total Knee Arthroplasty (THA/TKA). Methods: Eligibility Criteria: This study included English language original research papers from the USA, encompassing various study designs. Exclusion criteria comprise studies lacking empirical evidence on hospital financial performance. Information Sources: A comprehensive search using relevant keywords was conducted across databases from January 1990 to December 2019 (updated in March 2021), covering peer-reviewed articles and gray literature. Risk of Bias: Bias in the included studies was assessed considering study design, adjustment for confounding factors, and potential effect modifiers. Synthesis of Results: The review adhered to PRISMA guidelines. Employing Monte Carlo simulations, a meta-analysis was conducted with 100,000 simulated samples. Results indicated mean 30-day readmission costs: USD 16,037.08 (95% CI, USD 15,196.01–16,870.06) overall, USD 6852.97 (95% CI, USD 6684.44–7021.08) for AMI, USD 9817.42 (95% CI, USD 9575.82–10,060.43) for HF, and USD 21,346.50 (95% CI, USD 20,818.14–21,871.85) for THA/TKA. Discussion: Despite the financial challenges that hospitals face due to the ACA and the Hospital Readmissions Reduction Program, this meta-analysis contributes valuable insights into the consistent cost trends associated with 30-day readmissions. Conclusions: This systematic review and meta-analysis provide comprehensive insights into the financial implications of 30-day readmissions for specific medical conditions, enhancing our understanding of the nexus between healthcare quality and financial performance.

## 1. Introduction

The Patient Protection and Affordable Care Act of 2010, which is a federal law, has the primary purposes of expanding the health insurance coverage of US citizens and improving the quality of healthcare delivery. However, it presents financial threats and opportunities to many hospitals across the USA [1]. The enactment of the Patient Protection and Affordable Care Act (ACA) led to the implementation of initiatives such as the Hospital Readmissions Reduction Program (HRRP) to enhance healthcare quality through reducing hospital readmissions. Hospitals reimbursed under the Inpatient Prospective Payment System (IPPS) are expected to provide more and more predefined quality performance indicators. These changes in how hospitals are reimbursed can cause some hospitals to undergo financial sanctions [1,2]. The widespread expressions of chronic illnesses in the baby boomer generation translate into increasing demand for medical services. Hospitals face financial strains as the demand for quality services increases with higher patient expectations, coupled with penalties from the Centers for Medicaid & Medicare Services (CMS) Hospital Readmissions Reduction Program (HRRP) initiatives [3,4]. Historically, through the HRRP program, the CMS can withhold up to 3% of reimbursements for readmissions within 30 days, which exceeds national standards. The use of national standards is being criticized for not accommodating the case mix of patients [5,6]. Thus, active 2019 hospitals are ranked in quintiles depending on the proportion of dual-eligible patients that each hospital serves. Therefore, the new methodology will compare each hospital to the median readmission rates of its cohorts. It is unknown how this new methodology will affect the cost of readmission. 

### 1.1. New Contribution 

Despite the increasing dependence on 30-day readmission rates and patients’ case mix in determining hospital reimbursements, there seems to be limited research on how payment based on 30-day readmission is related to the cost of readmissions. This lack of attention is evident because no meta-analysis on the relationship between financial performance and 30-day readmission rates has been published. To our knowledge, this is the only meta-analysis of the literature to date exploring the relationship between 30-day readmission rates and costs. Given that meta-analysis is considered the gold standard in analyzing, synthesizing, and integrating available literature on quality and financial outcomes [7], this study has significant potential for informing future research. A synthesis of the novel literature is included in this study to account for recent trends on how hospitals perform financially on the indicators of 30-day readmission rates, as this is needed to streamline our meta-analysis and guide future studies.

This study aims to contribute to the literature by adding more knowledge on previous studies [8,9] by specifically looking at studies including the quality variables of 30-day readmission rates and financial performance, which were not included in previous studies. A focus on this neglected quality aspect will allow for more inferences to be drawn about salient variables in hospital settings implementing the HRRP.

### 1.2. Conceptual Framework

This study adopted the quality–cost framework using the Donabedian Model of structure, process, and outcomes [8,9,10,11,12]. Our quality measure is the reduction in the cost of 30-day readmission rates. The structure represents the resources that hospitals use to reduce 30-day readmissions, like increasing staff ratios, equipment, and institutional/treatment protocol [13,14]. The structure also encompasses the formal and informal systems through which healthcare is financed, like the insurance structure, healthcare policies, healthcare worker availability, and available healthcare delivery systems [15,16] Within the conceptual framework’s process dimension, we delve into the dynamic elements of patient care: assessments, examinations, and a spectrum of healthcare interactions. This expansive perspective recognizes that the efficacy of these processes is intricately linked to the overarching cost dynamics of readmissions. Our study underscores the critical role that various processes play in influencing healthcare expenditure, emphasizing the need to scrutinize and optimize these aspects to mitigate the costs associated with 30-day readmissions. By unpacking the multifaceted process dimension, our research aims to provide a robust understanding of how healthcare processes contribute to the overall financial landscape in the context of readmissions [15]. This comprehensive perspective recognizes that the effectiveness of these processes is intricately linked to the overarching cost dynamics of 30-day readmissions [16].

We carried out a literature review of how hospital readmissions affect financial performance and located the readmission cost as our variable of interest for the meta-analysis rooted in the rationale provided by the quality–cost framework. This framework delineates the relationship between the quality aspects of the cost of readmission in hospitals. It maps the structure and determines the process, ultimately leading to outcomes. For the structure, we consider the quality improvement measures for the subdomains of quality-specific diseases of 30-day readmission for Acute Myocardial Infarction (AMI), Chronic Obstructive Pulmonary Disease (COPD), Heart Failure (HF), Pneumonia, Coronary Artery Bypass Graft (CABG) surgery, Elective Primary Total Hip Arthroplasty and/or Total Knee Arthroplasty (THA/TKA), as well as quality improvement measures [17].

Within this framework (Figure 1), we delineate the structure to include quality improvement measures to reduce readmissions for AMI, COPD, HF, Pneumonia, CABG Surgery, and THA/TKA. The process includes subdomains related to medical errors and appropriate care that affects readmissions, while outcomes involve disease progression and care complications for various medical conditions. Considering that each of these quality attributes inherently has cost implications for monitoring and evaluation and subsequently influences healthcare costs, the conceptual foundation is built upon the foundational work of previous studies [18,19] which identified the dimensions of profitability: profitability, liquidity, capital structure, activity, cost, revenue, and utilization. Conducting a thorough systematic review that reveals other financial performance measures was crucial to understand the research landscape in this domain and to justify why cost was isolated for the meta-analysis. We derived the following research question building on this framework. 

What is the relationship between financial performance variables reported as independent factors for 30-day readmission, the different pathological conditions associated with 30-readmission, and the significant findings derived from the average cost of readmission?

Meta-analysis precedes the literature review to comprehensively understand the interplay between financial aspects and readmission. This approach enables us to glean insights into the intricate relationship between quality improvement, financial performance measures, and the cost measure of financial performance with hospital readmission. 

## 2. Methods 

We carried out a meta-analysis on the Covidence software following the Preferred Reporting Items for Systematic Review and Meta-Analysis (PRISMA). A three-step procedure was used for the review. The keywords for financial performance, 30-day readmission rates, and the hospital setting, as defined by the Boolean operators OR/AND, were used to arrange the keywords (Figure 2). The search strategy used quality, financial performance headings and keywords and their combinations “30-day readmission rates”, “patient readmission”, “re-hospitalization”, “reoperation”, AND/OR “hospital” “acute care”, “ acute care hospital” “critical care” AND/OR “cost”, “revenue”, “profitability”, “total margin”, “operating margin”, “return on investment”, “financial performance”, “financial”, “accounting”, “financing”, “activity”, and “outcome.” Peer-reviewed articles were located using the following databases: ABI/INFORM, Web of Science, Scopus, PubMed/MEDLINE Medline, Embase, and Academic Search Premier.

This meta-analysis used PRISMA guidelines (Figure 3). We considered the following types of studies for inclusion: full original research papers written in the English language, randomized or non-randomized controlled trials, prospective or retrospective cohort studies, cross-sectional studies, pilot studies, and studies from the USA. For optimal search outcomes, we extracted peer-reviewed articles, gray literature written only in English, published between January 1990 and December 2019, and updated in March 2021 and February 2024. The effects described were proven change or no change in financial performance for 30 days of readmission. We excluded the studies that did not include the hospital financial performance as the outcome or comparator and qualitative studies without empirical evidence for hospital financial performance. Abstracts were screened for studies not carried out within the United States to account for variations in international epidemiologic, economic, and medical practice. 

### 2.1. Outcomes

We extracted the independent variables of 30-day readmission rates as the dependent variable. We calculated the inflation-adjusted financial performance using the standardized abstraction protocol and the Covidence tool by three abstractors. Financial performance outcomes included cash flow margin, charges, income, cost revenue, operating margin, return on investment, operating expenses, operating revenue, return on assets, total margin, operating expense, and one-year subsequent Medicare spending. We analyzed studies conducted with the patient populations of hospitals and hospital populations separately to reduce any bias related to hospitals’ financial performance. A separate analysis was conducted for all-cause readmission rates, acute myocadiac infarction, heart failures, and other illnesses. The studies were analyzed for design and adjustment for confounding factors for possible effect modifiers. Various authors have used different units of measurement for financial performance. We adapted to use articles reporting financial performance as a measure of the USD value and eliminated articles that only reported quality aspects.

We took a second secondary review of abstracted publications from Covidence to determine whether articles on the borderline met the inclusion criteria. This process illuminated 11 articles primarily linked to the methods of reporting economic outcomes. We identified 24 studies estimating the attributable cost of all-cause 30-day readmission. Within the included articles, minimal variations in methods, data sources, and settings could not be avoided. All included articles generated average attributable costs from readmission. In cases where other variables, in addition to cost, were reported, we only considered the cost and charge component. Using the charge, we estimated the cost using a cost-to-charge ratio of 0.50, as used by [1,2].

To bolster the methodological rigor and align more closely with PRISMA guidelines, studies that reported the effect measures for each outcome were explicitly stated, specifying the metric employed, be it risk ratio, mean difference, or other relevant measures. In addressing heterogeneity, our discussions elucidate the methods applied, such as subgroup analyses or meta-regression, to explore potential variations among the study results comprehensively. To tackle the reporting bias, we employed Monte Carlo simulations to assess and mitigate the biases arising from missing results. These refinements enhance the transparency and thoroughness of the meta-analysis, fostering a more robust adherence to PRISMA guidelines and affording readers a nuanced insight into the methodologies underpinning this systematic review. Our study sought to estimate the mean cost of 30-day readmissions, a probabilistic outcome subject to uncertainty due to various factors such as patient characteristics, treatment effectiveness, and healthcare processes. Monte Carlo simulation is well suited for handling such uncertainty by repeatedly sampling from input parameter distributions to estimate the distribution of possible outcomes. In this case, the simulation would allow for the assessment of the uncertainty around the mean cost estimates and provide confidence intervals.

### 2.2. Statistical Analysis

The results reported in the articles were very heterogeneous. For example, different financial performances were reported (Marginal Cost, Incremental Cost, Operating Revenues, Operating Expenses, and Operating Margin). Operating Margin, Cost, and Return on Investment) with a specific lens on readmission costs. The reporting of cost also differed across papers as some reported raw USD values while others reported mean/median costs. In addition, different pathological conditions were reported. Most of the studies did not mention controls (ideally 30-day unadjusted mortality rates), making it exceedingly difficult to perform a meta-analysis.

The literature review results provided average estimates of the cost of a 30-day readmission. For each study included, a weighted average of the point estimate was used to calculate the cost estimate relative to the sample size. To assess the consistency of the association between 30-day readmission rates and financial performance outcomes across several studies, we conducted a Monte Carlo simulation to develop confidence intervals (CI) for every point estimate. We achieved this by generating our data to see the trend and creating an estimator to see how close we are to the trend. We analyzed the studies separately, considering the data years to adjust for inflation. Considering that our data came from various sources, we chose Monte Carlo simulations because of their ability to realistically estimate uncertainty.

Meta-analysis was chosen as the primary research method to synthesize data from multiple studies and provide a robust estimation of the mean cost of 30-day readmissions across various medical conditions. This method allows for the integration of findings from disparate studies to derive more precise and generalizable conclusions.

The meta-analysis process involved several key steps to ensure methodological rigor and validity using the meta-analysis flow chart depicted in Figure 4. 

To develop the confidence interval through Monte Carlo simulations, we conducted a series of sensitivity analyses using the variant approach suggested in the research [20,21] for health service research cost estimates. For each study’s probabilistic distribution of a cost estimate, a Monte Carlo simulation was conducted with 100,000 trials. The output was a triangular and general distribution with a low end, most probable point estimate, and a high end.

## 3. Results 

The articles from the literature search were carefully reviewed based on inclusion/exclusion criteria. A total of 38 studies were found in the systematic review to be eligible for inclusion and were considered for further analysis (Table A1). Studies were further categorized into five different categories (Acute Myocardial Infarction (AMI), Chronic Obstructive Pulmonary Disease (COPD), Heart Failure (HF), Pneumonia, Coronary Artery Bypass Graft (CABG) surgery, and Elective Primary Total Hip Arthroplasty and/or Total Knee Arthroplasty (THA/TKA)) based on the disease condition. 

A Monte Carlo simulation method was carried out to estimate the average cost of 30-day readmission conditions due to its capacity to provide reasonably accurate uncertainty forecasts. For the Monte Carlo analysis, first, we created the confidence intervals for the 30-day readmission cost reported in this article. Then, the identified 38 studies providing the reasonable cost estimates of the attributable cost of readmissions (Figure 4). We identified 4 studies for AMI, 6 studies for HF, 6 studies for THA/TKA, and 22 studies for all other readmissions. 

We simulated the distribution for each suitable analysis before pooling the results and weighting the results by sample size. We followed the method described another meta-analysis simulating cost [21,22], which is based on three observations: point figures for the three experiments that make up the strongest indicator of central inclination, as well as lower and upper limits. For each readmission reported in the study, we used the 95 percent CI to set the endpoints for the distribution if it were either stated in the article or could be estimated so that 2.5 percent of the distribution falls below the lower and above the upper value. We then tested to see whether the modeled triangular distribution accurately matched the study results by setting the most possible value of the triangular distribution equal to the reported central propensity scale. 

Finally, we conducted Monte Carlo simulations using @RISK software to analyze the data. Specifically, we simultaneously simulated 100,000 sample draws from the modeled distribution for each related analysis across all readmission categories. At each iteration, we determined the weighted average of the included studies. We recorded the mean and 95 percent confidence interval obtained from the distribution of those 100,000 weighted averages for each readmission category. Subsequently, we calculated the mean cost of 30-day readmission for all conditions and specific conditions, such as AMI (Figure 5), HF (Figure 6), and THA/TKA (Figure 7), along with their respective confidence intervals. Monte Carlo simulations were carried out with the help of the Monte Carlo simulation software @RISK, version 7.6.1. (Palisade Corp., Ithaca, NY, USA). Following the Monte Carlo interactions seen in (Figure 8), the mean cost of 30 days readmission for all conditions is simulated at USD 16,037.08 (95% CI, USD 15,196.01–16,870.06). The mean cost of 30-day readmission for AMI is USD 6852.97 (95% CI, USD 6684.44–7021.08). The mean cost of 30-day readmission for HF is estimated at USD 9817.42 (95% CI, USD 9575.82–10,060.43). The mean cost of 30 days readmission for THA/TKA is simulated at USD 21,346.50 (95% CI, USD 20,818.14–21,871.85)”.

## 4. Discussion

Two decades after the landmark article “To Err is Human” by the Institute of Medicine (IOM), patient safety and quality improvements have been claimed to be at the forefront of many initiatives. The CMS implemented the readmission reduction program to reduce 30-day readmissions and improve quality-of-care efforts that can lead to significant cost reduction [23]. In this study, we conducted a systematic review and meta-analyses to explore the financial implications of a 30-day readmission reduction program. To achieve our goal, for the six pre-standardized unplanned readmission measures, following our search criteria, we found articles for AMI, HF, and THA/TKA that reported the attributable cost of readmission. We did not include readmissions for COPD, pneumonia, and CABG, as these conditions were not included in the 38 articles retained for this meta-analysis. These results were unexpected as most research has been performed on AMI, HF, and pneumonia [24,25]. This is true because the HRRP was established in 2010 with the initial target indicators AMI, HF, and pneumonia [2,26]. It is, therefore, surprising that the studies we found reposrted limited the attributable costs of pneumonia that meet our inclusion criteria.

In this section, we seek to bolster the robustness and credibility of our Monte Carlo simulation estimates by aligning them with real-world data from the Healthcare Cost and Utilization Project (HCUP) for the year 2018. This comparative analysis serves as a pivotal step in validating our findings. By referencing the HCUP data, we corroborate the accuracy of our Monte Carlo simulations, enhancing their overall credibility and robustness. This approach validates our estimates and significantly contributes to our comprehension of hospital readmissions, their associated costs, and their far-reaching impacts on healthcare quality and financial performance.

Our study estimated the average cost of 30-day all-cause adult hospital readmissions at USD 16,037.08. This estimate closely aligns with the data reported in HCUP, which indicated an average readmission cost of USD 15,200 for the same period. This consistency strengthens the validity of our estimates (HCUP, 2018). The average cost of readmission that we found through the meta-analysis was USD 16,870.06 (95% CI, USD 15,196.01–16,870.06). This amount is above the readmission rates reported by [27,28] for rural community hospitals at USD 2683 and USD 2248.21, respectively. On the other hand, the subacute hospitals had a significantly higher cost of readmission USD 15,563 [27]. These results indicate the geographical differences in the cost of readmission to build sustainable healthcare systems with unwarranted variety in the quality of care and cost. These challenges should be avoided. It is crucial to understand the geographical distribution of unplanned readmissions and how these variations impact the cost of readmissions [28].

In line with our study’s findings, the Department of Veteran Affairs HERC Health Economics Seminar, specifically Jason Hockenberry’s presentation on ‘The Cost of Readmissions: Implications for Reimbursement Policies,’ highlights the challenge of establishing a substantial relationship between hospital readmission rates and costs. Hockenberry’s observations indicate that the coefficient on the hospital readmission rate from the previous period is notably small and statistically insignificant. This aligns with the complex nature of the financial implications of readmissions, where even a negative cost effect of USD 12.00–USD 31.00 is deemed a minor impact (Department of Veteran Affairs HERC Health Economics Seminar, 2018).”

In line with other studies, our results indicated that among the three 30-day risk-standardized unplanned readmission measures reported, THA/TKA reported the highest attributable cost per readmission, USD 21,346.50 (95% CI, USD 20,818.14–21,871.85) [29,30,31,32]. These results align with the CMS adding THA/TKA to the HRRP because of the high prevalence, its increased number of readmissions, and the high overall Medicare expense for this measure [33].

The results of our estimates suggest questions on the unintended cost implications of the HRRP with the various five measures grouped together and individually. First, within our scope, this study is the first meta-analysis to simulate the cost of readmissions after the onset of the HRRP. Despite the lack of experimental studies, observed variations in the cost of readmission provide evidence that the CMSs were right to expand the HRRP measures. More financial performance studies are warranted to inform policy since the HRRP was designed to improve the quality by reducing hospital readmissions and decreasing CMS spending [34].

### Limitations

Our research encountered several limitations. Firstly, the inclusion of different pathological conditions in the studies created challenges in consolidating the data. Secondly, most studies lacked the mention of controls, specifically 30-day unadjusted mortality rates, which posed significant obstacles to conducting a meta-analysis. Thirdly, the cost estimates derived from peer-reviewed articles exhibited considerable heterogeneity. Authors presented various types of costs and financial performance measures, with discrepancies in cost reporting methods. Despite our efforts to mitigate these variations through a rigorous review process, we primarily focused on mean cost estimates for inclusion. Indeed, for certain readmission measures, we excluded some articles due to the extent of heterogeneity observed. To address the inherent uncertainty and heterogeneity in our data, we employed Monte Carlo simulations as part of our analysis. There is a clear need for robust studies to comprehensively assess the cost of readmissions.

There is the possibility that our results contain some underestimations resulting from the population under study. The study is limited to an adult population readmitted to acute care hospitals. We excluded non-acute, long-term care facilities, and pediatric acute hospitals. The total attributable costs for readmission for the entire US healthcare system are most likely higher, warranting an increase in readmission reduction initiatives. Finally, we acknowledge the presence of comorbidities that can impact the average readmission cost. Although we attempted to account for co-morbidities and primary diagnosis in our included studies, it is possible that this might not be a complete list. 

## 5. Conclusions

Our study offers valuable insights into the financial burden imposed by re-admissions on acute care hospitals, providing robust estimates of attributable cost resources for readmissions across various medical conditions. By quantifying the mean costs of 30-day readmissions for conditions such as AMI, HF, and THA/TKA, our findings shed light on the economic impact of readmissions and highlight areas where healthcare resources are being allocated.

One of the primary benefits of our study is its provision of concrete estimates that can inform decision-making and resource allocation strategies for healthcare stakeholders, including hospital administrators, policymakers, and payers. By understanding the financial implications of readmissions, stakeholders can develop targeted interventions and quality improvement initiatives to reduce readmission rates and optimize healthcare spending.

Furthermore, our study opens avenues for future research by identifying persistent trends in readmission costs and emphasizing the need for continued efforts to address this challenge. Future studies can build upon our findings by investigating the effectiveness of specific interventions and strategies to reduce readmissions and improve overall healthcare outcomes. Additionally, exploring the impact of demographic and clinical factors on readmission costs could provide further insights into the drivers of healthcare expenditure.

In light of the ongoing emphasis on value-based care and healthcare cost containment, our study underscores the importance of addressing readmissions as a critical component of healthcare quality improvement efforts. By addressing the financial implications of readmissions, hospitals can better align their resources and interventions to improve patient outcomes while optimizing healthcare spending.

## Figures and Tables

**Figure 1 healthcare-12-00750-f001:**
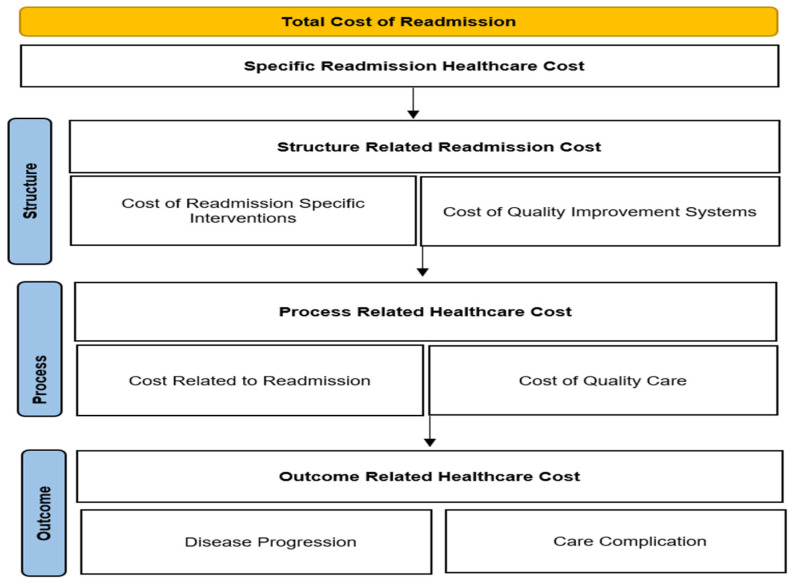
Cost that can be influenced by quality components of readmission. Adopted from [12].

**Figure 2 healthcare-12-00750-f002:**
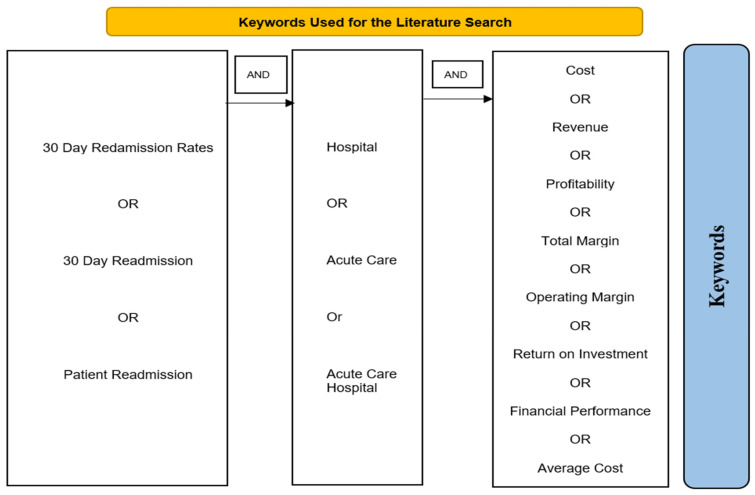
Keywords used for the literature search.

**Figure 3 healthcare-12-00750-f003:**
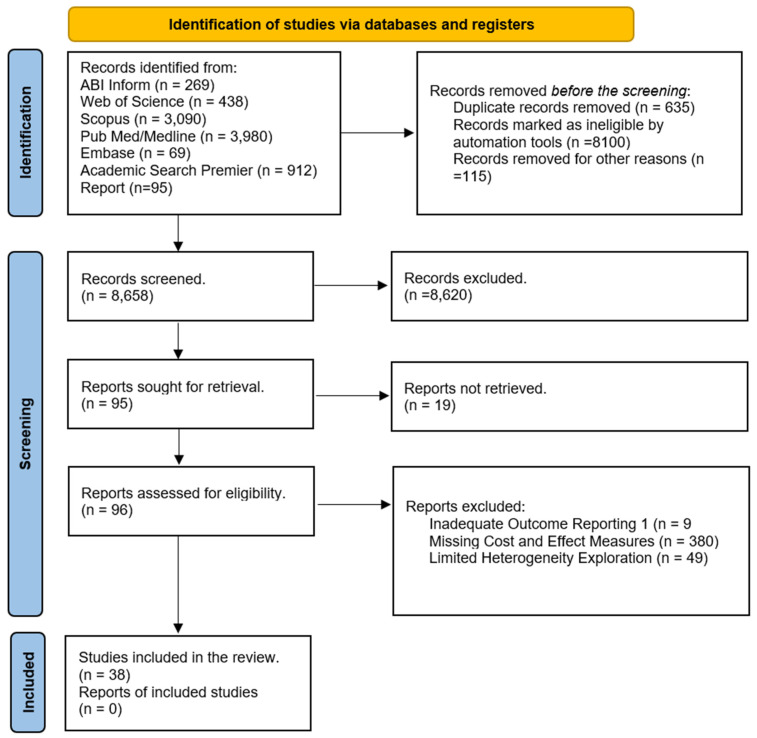
Cohort selection for systematic review and meta-analysis.

**Figure 4 healthcare-12-00750-f004:**
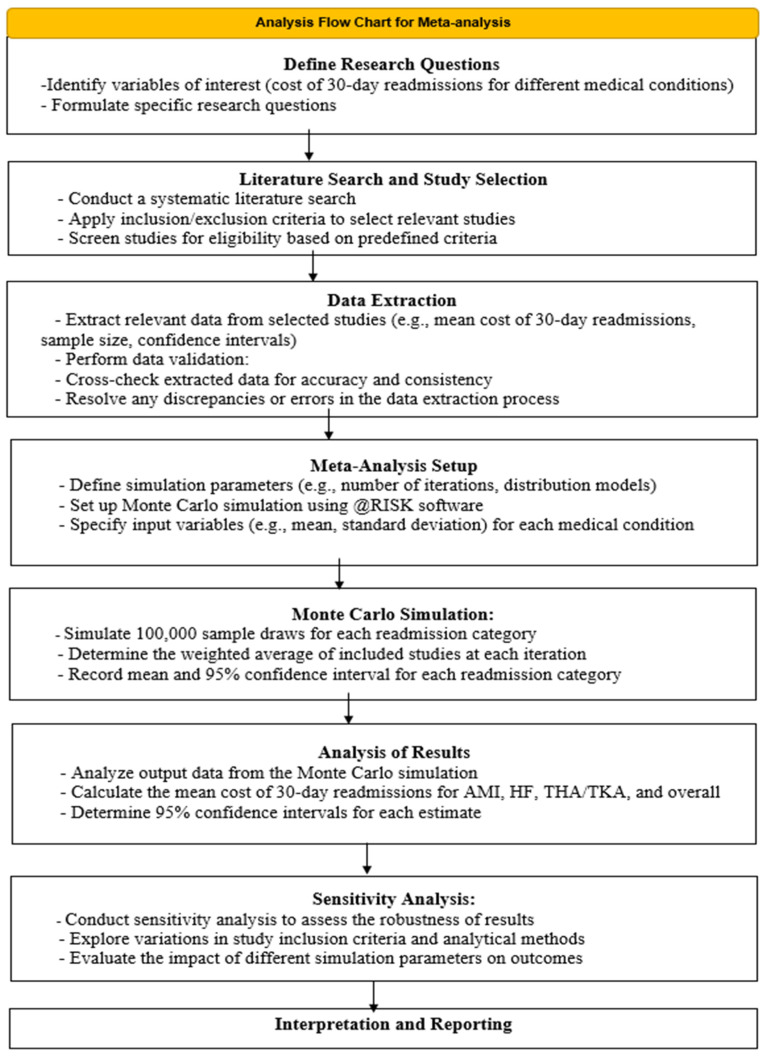
Analysis flow chart for meta-analysis.

**Figure 5 healthcare-12-00750-f005:**
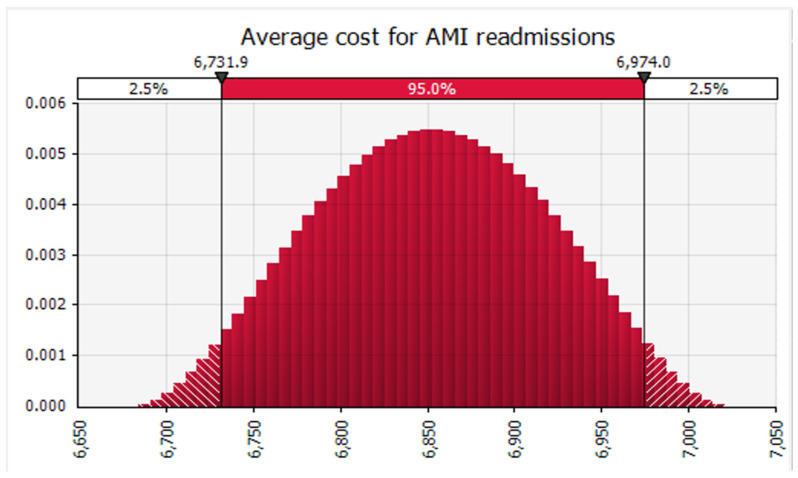
Average cost of 30 Day Readmission for Acute Myocardial Infarction.

**Figure 6 healthcare-12-00750-f006:**
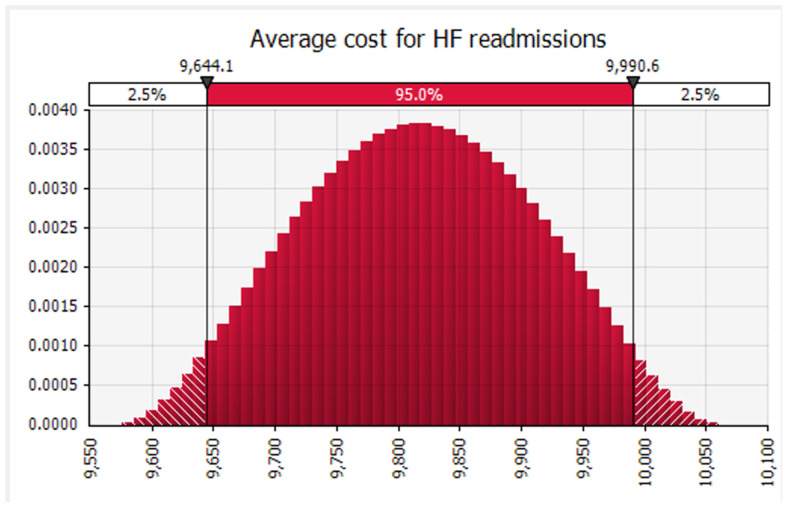
Average cost of 30 Day Readmission for Heart Failure.

**Figure 7 healthcare-12-00750-f007:**
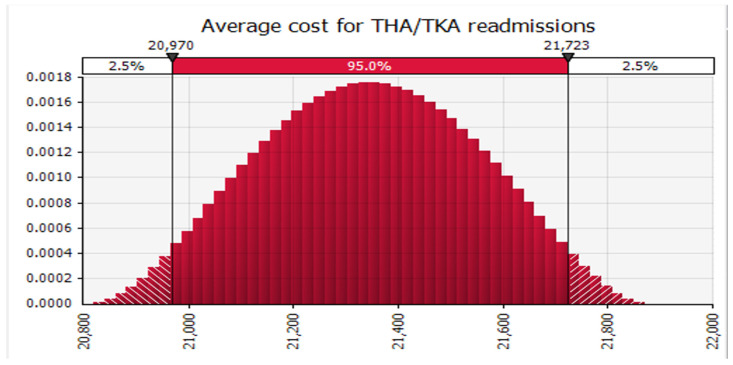
Average Cost of 30 Day Readmission for Total Hip Arthroplasty (THA) and Total Knee Arthroplasty (TKA) procedures.

**Figure 8 healthcare-12-00750-f008:**
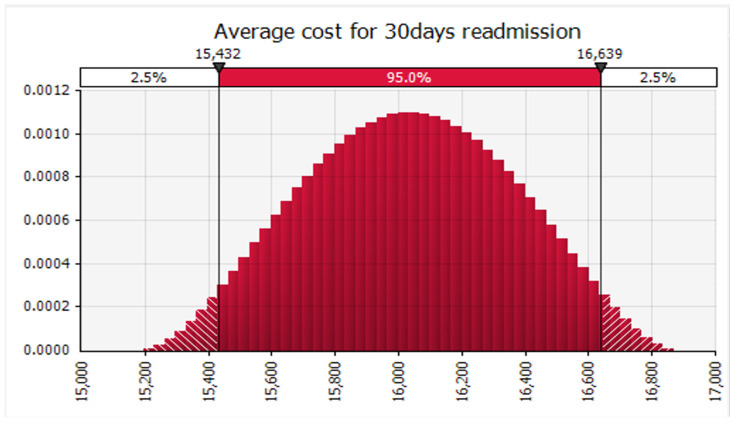
Average cost of 30 Day readmission.

## Data Availability

Data are contained within the article.

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
