# Peer review of "Systematic Review and Meta-Analysis of the Financial Impact of 30-Day Readmissions for Selected Medical Conditions: A Focus on Hospital Quality Performance"

_healthcare, 2024, doi:10.3390/healthcare12070750_

Round 1
Reviewer 1 Report
Comments and Suggestions for Authors
In this study, the authors conducted a systematic literature search on the financial effects of a 30-day readmission and tried to calculate the average cost. Obtaining peer review articles through a systematic literature search increased the quality of the study. The following situations need to be taken into account:
1. The Patient Protection and Affordable Care Act (ACA) institution was mentioned in the abstract. Where and by whom is this institution managed?
2. It is quite appropriate to include the crucial results of the study and to mention the source of motivation and purpose of the study in the abstract, but it is quite long. Therefore it should be shortened a bit?
3. There are grammatical and spelling errors. Please check throughout the article.
4. Explain why you chose to use Monte Carlo Simulation in the study. Discrete event simulation method is also used to develop a confidence interval. It can even take stochastic behavior into account better than the Monte Carlo Simulation method. Why did you use the Monte Carlo Simulation method instead of the Discrete Event Simulation method?
5. The Conclusion section is very inadequate. Mention the benefits of this study to its stakeholders.
6. In the Conclusion section, discuss the opportunities of the study for future studies in more detail.
Comments on the Quality of English LanguageMinor editing of English language required.
Author Response
|
|
Reviewer #1 comments |
Authors’ response |
|
1. |
It is quite appropriate to include the crucial results of the study and to mention the source of motivation and purpose of the study in the abstract, but it is quite long. Therefore, it should be shortened a bit? |
I We really appreciate the reviewers’ feedback for improving the manuscript. The revised manuscript incorporates substantial revisions to the abstract and conclusion that seek to address the concerns expressed by Reviewer 1. “The length of the abstract has been reduced from 420 words to 281 words” |
|
2. |
2. In this study, the authors conducted a systematic literature search on the financial effects of a 30-day readmission and tried to calculate the average cost. Obtaining peer review articles through a systematic literature search increased the quality of the study. The following situations need to be taken into account: i. The Patient Protection and Affordable Care Act (ACA) institution was mentioned in the abstract. Where and by whom is this institution managed? ii. There are grammatical and spelling errors. Please check throughout the article. iii. Minor editing of English language required. |
This concern is addressed throughout the article. “The Patient Protection and Affordable Care Act (ACA) was signed into law by President Barack Obama in 2010” |
|
3 |
Explain why you chose to use Monte Carlo Simulation in the study. Discrete event simulation method is also used to develop a confidence interval. It can even take stochastic behavior into account better than the Monte Carlo Simulation method. Why did you use the Monte Carlo Simulation method instead of the Discrete Event Simulation method? |
We appreciate the reviewers’ feedback for improving the manuscript. The revised manuscript incorporates substantial revisions to the Methods and Conclusion that seek to address the concerns expressed by Reviewer 1. This comment is addressed in lines 1973-200 in the manuscript. “Our study sought to estimate the mean cost of 30-day readmissions, a probabilistic outcome subject to uncertainty due to various factors such as patient characteristics, treatment effectiveness, and healthcare processes. Monte Carlo simulation is well-suited for handling such uncertainty by repeatedly sampling from input parameter distributions to estimate the distribution of possible outcomes. In this case, the simulation would allow for the assessment of the uncertainty around the mean cost estimates and provide confidence intervals.”
"In addition to the aforementioned reasons, Discreet Event Simulation has increasingly been applied to evaluate specific technologies in health technology assessment and we did not find studies that used decreet event simulation to address the cost of quality of care.
https://www.ncbi.nlm.nih.gov/pmc/articles/PMC5746244/ https://journals.sagepub.com/doi/full/10.1177/0272989X12455462
https://www.sciencedirect.com/science/article/pii/S0196655319309605
It is noteworthy that Monte Carlo simulation has been widely utilized in the literature concerning meta-analyses of costs and financial impacts. This approach is prevalent due to its robustness in handling the inherent uncertainty and variability in financial data and cost estimations. By repeatedly sampling from input parameter distributions, Monte Carlo simulation allows researchers to effectively capture and quantify potential outcomes, providing more comprehensive insights into the financial implications being studied. Therefore, the choice of Monte Carlo simulation in the study aligns with established practices in meta-analyses of costs and economic impacts, ensuring methodological consistency and reliability in estimating the mean cost of 30-day readmissions for different medical conditions. Monte Carlo Simulation: https://www.cambridge.org/core/journals/infection-control-and-hospital-epidemiology/article/modeling-the-costs-of-hospitalacquired-infections-in-new-zealand/8680C4144310043D712A6FEC8582DF23 https://www.tandfonline.com/doi/abs/10.1057/jos.2009.17 https://link.springer.com/article/10.1186/1471-2334-10-247
Discrete Event Simulation (DES): While DES could potentially model the progression of diseases or treatment pathways in healthcare, it may not be as applicable in this context. The study primarily focused on estimating costs rather than modeling dynamic processes with discrete events. The outcome of interest, the cost of 30-day readmissions, is not inherently tied to specific events occurring at distinct points in time, but rather to the overall probabilistic nature of healthcare costs." |
|
4 |
The Conclusion section is very inadequate. Mention the benefits of this study to its stakeholders. |
This comment is addressed in lines 367-390 in the manuscript. “Our study offers valuable insights into the financial burden imposed by re-admissions on acute care hospitals, providing robust estimates of attributable cost resources for readmissions across various medical conditions. By quantifying the mean costs of 30-day readmissions for conditions such as AMI, HF, and THA/TKA, our findings shed light on the economic impact of readmissions and highlight areas where healthcare resources are being allocated. One of the primary benefits of our study is its provision of concrete estimates that can inform decision-making and resource allocation strategies for healthcare stakeholders, including hospital administrators, policymakers, and payers. By understanding the financial implications of readmissions, stakeholders can develop targeted interventions and quality improvement initiatives to reduce readmission rates and optimize healthcare spending. Furthermore, our study opens avenues for future research by identifying persistent trends in readmission costs and emphasizing the need for continued efforts to address this challenge. Future studies can build upon our findings by investigating the effectiveness of specific interventions and strategies to reduce readmissions and improve overall healthcare outcomes. Additionally, exploring the impact of demographic and clinical factors on readmission costs could provide further insights into the drivers of healthcare expenditure. In light of the ongoing emphasis on value-based care and healthcare cost containment, our study underscores the importance of addressing readmissions as a critical component of healthcare quality improvement efforts. By addressing the financial implications of readmissions, hospitals can better align their resources and interventions to improve patient outcomes while optimizing healthcare spending.” |
Reviewer 2 Report
Comments and Suggestions for Authors
This study aims to systematically review the literature and conduct a meta-analysis to estimate the average cost of procedure-specific 30-day risk-standardized unplanned readmissions. This study raises a valuable academic topic, and adopts some mainstream research methods, which have reference significance for future research. However, the authors need to focus on the following issues and consider further revisions.
1. Further clarification should be added on the research question. Although this article proposes three research questions, the logical connection between these three questions is not specified and could be merged into one research question. In addition, the three questions raised in this article are more indicative of the research objectives of the authors, indicating the specific information obtained from existing research. If the exploration of the information is used as research question, it may significantly reduce the innovation of this article.
2. This article proposes the use of meta-analysis as a research method, but there is too little introduction for this method. According to the current analysis in the article, it seems that the analysis process does not comply with the general norms of meta-analysis and lacks important steps such as data validation. The data results of meta-analysis appear too simplistic, resulting in a lack of data support for the formed research conclusions. I suggest providing an analysis flowchart for meta-analysis and explaining the main content and interrelationships of each analysis module. In addition, it is recommended that authors cite more papers using meta-analysis methods recently to enhance the depth and reliability of the results.
3. There should be a lot of improvements for the charts and tables in this article. There are many charts that appear non-standard, requiring optimization of element layout and overall structure, and also improving the clarity of the charts. I recommend that the style of Figures 1 and 2 could be consistent with Figure 3. Table 1 takes up too much space, it is recommended to include it in the appendix or supplementary materials.
Author Response
|
|
Reviewer #2 comments |
Author's response |
|
1. |
Further clarification should be added on the research question. Although this article proposes three research questions, the logical connection between these three questions is not specified and could be merged into one research question. In addition, the three questions raised in this article are more indicative of the research objectives of the authors, indicating the specific information obtained from existing research. If the exploration of the information is used as research question, it may significantly reduce the innovation of this article |
We appreciate the reviewers’ feedback for improving the manuscript. The revised manuscript incorporates substantial revisions to the conceptual framework, methods, and conclusion that seek to address the concerns expressed by Reviewer 2
To address this concern, we have refined and merged the three proposed questions into an overarching research question. This consolidated question will encapsulate the essence of the study objectives while maintaining clarity and focus (Page 7&8, Lines 122-124):
“What is the relationship between financial performance variables reported as independent factors for 30 day readmission, the different pathological conditions associated with 30 day readmission, and the significant findings derived from the average cost of readmission?” |
|
2. |
This article proposes the use of meta-analysis as a research method, but there is too little introduction for this method. According to the current analysis in the article, it seems that the analysis process does not comply with the general norms of meta-analysis and lacks important steps such as data validation. The data results of meta-analysis appear too simplistic, resulting in a lack of data support for the formed research conclusions. I suggest providing an analysis flowchart for meta-analysis and explaining the main content and interrelationships of each analysis module. In addition, it is recommended that authors cite more papers using meta-analysis methods recently to enhance the depth and reliability of the results. |
We have added an analysis flowchart for meta-analysis and explaining the main content and interrelationships of each analysis module and improved paragraph 3 of the Statistical Analysis section. Pg 8, Lines 220-225):
Meta-analysis was chosen as the primary research method to synthesize data from multiple studies and provide a robust estimation of the mean cost of 30-day readmissions across various medical conditions. This method allows for the integration of findings from disparate studies to derive more precise and generalizable conclusions. The meta-analysis process involved several key steps to ensure methodological rigor and validity using the meta-analysis flow chart depicted in Figure 4.
|
|
3. |
There should be a lot of improvements for the charts and tables in this article. There are many charts that appear non-standard, requiring optimization of element layout and overall structure, and also improving the clarity of the charts. I recommend that the style of Figures 1 and 2 could be consistent with Figure 3. Table 1 takes up too much space, it is recommended to include it in the appendix or supplementary materials. |
We appreciate the feedback regarding the layout and structure of our article's charts and tables. We have considered your suggestions and made the necessary improvements. We have adjusted the layout structure of Figures 1, 2 and 4 to align with the style of Figure 3, ensuring consistency throughout the text. Additionally, we have addressed the issue of Table 1 occupying excessive space by relocating it to the Appendix. These revisions aim to enhance the clarity and readability of the visual elements in our article. Thank you for your valuable input, and we believe these changes will significantly improve the overall presentation of our research findings. |
Reviewer 3 Report
Comments and Suggestions for Authors
ABSTRACT:
1. In the abstract you have the methods in bold and then the discussion. The results are not highlighted, please change this.
2. ACA (Affordable Care Act) is a federal law aimed primarily at influencing American citizens who do not have any health insurance. Although your work does not pertain to this topic, it is worth clarifying in the introduction.
3. Why the search covered peer-reviewed articles and gray literature published between January 29 1990 and December 2019, with an update in March 2021? These are quite old data. Why haven't they been extended, for example, until 2023?
4. The meta-analysis, 32 following PRISMA guidelines, involved simulating 100,000 samples for each related analysis. - I don't understand what samples - please explain.
5. Why you choose Monte Carlo symulation?
BACKGROUND
1. Within this framework, we delineate the structure to include quality improvement measures to reduce readmission - You do not mention this at all in the conclusions, so how are you providing means for quality improvement to reduce readmissions?
RESULTS
1. Table 1 is stretched over several pages, making it completely unreadable. I believe your task should be to present the results of the meta-analysis in a focused manner on key elements. Certainly, the construction of Table 1 needs reconsideration, and it might be proposed in a different form. Perhaps, some parts could be described?
2. Figure 4 is unclear.
3. this part is most important, you shoul describre more details: Finally, we simulated 100,000 sample draws from the modeled distribution for each related analysis for each readmission category simultaneously. We determined the weighted average of the included studies at each iteration. For each readmission category, we recorded the mean and 95 percent confidence interval obtained from the distribution of those 100,000 weighted averages. Monte Carlo simulations were carried out with the help of the Monte Carlo simulation software @RISK, version 7.6.1. (Palisade Corp). Following the Monte-Carlo interactions, the mean cost of 30-days readmission for all conditions is simulated at $16,037.08 (95% CI, $15,196.01 – $16,870.06). The mean cost of 30-days readmission for AMI is $6,852.97 (95% CI, $6,684.44 – $7,021.08). The mean cost of 30-days readmission for HF is estimated at $9,817.42 (95% CI, $9,575.82 – $10,060.43). The mean cost of 30-days readmission for THA/TKA is simulated at $21,346.50 (95% CI, $20,818.14- $21,871.85).
CONCLUSION
1. The conclusions do not reference the results, are too general, and rather brief. They lack practical implications of the study.
Author Response
|
|
Reviewer #3 comments |
Authors’ response |
|
1. |
In the abstract you have the methods in bold and then the discussion. The results are not highlighted, please change this. |
Thank you for bringing this to our attention. We have revised the abstract to ensure consistency in highlighting the different sections. The Results section is also highlighted in bold, along with the Methods and Discussion sections. (Abstract Pg 1 line 15-37) |
|
2 |
ACA (Affordable Care Act) is a federal law aimed primarily at influencing American citizens who do not have any health insurance. Although your work does not pertain to this topic, it is worth clarifying in the introduction. |
Thank you for your suggestion regarding clarifying the Affordable Care Act (ACA) in the introduction. We agree that providing context on such a significant federal law can be beneficial for readers (Introduction Pg 1& 2 line 41-45) “The Patient Protection and Affordable Care Act of 2010, which is a federal law, has the primary purposes of expanding health insurance coverage among US citizens and improving the quality of healthcare delivery. However, it presents financial threats and opportunities to many hospitals across the USA [1]. The enactment of the Patient Protection and Affordable Care Act (ACA) led to implementing initiatives such as the Hospital Readmissions Reduction Program (HRRP) to enhance healthcare quality through reducing hospital readmissions.” |
|
3 |
Why the search covered peer-reviewed articles and gray literature published between January 29, 1990, and December 2019, with an update in March 2021? These are quite old data. Why haven't they been extended, for example, until 2023? |
Thank you for your input. While our initial search covered articles up to March 2021, we extended our review to February 2024 and found no new articles meeting our criteria. We'll consider extending the search period for future updates.
|
|
4 |
The meta-analysis, 32 following PRISMA guidelines, involved simulating 100,000 samples for each related analysis. - I don't understand what samples - please explain. |
Thank you for your feedback. The abstract has been revised and shortened for clarity (Results section Pg line 27 -31) “The review adhered to PRISMA guidelines. Employing Monte Carlo simulations, a meta-analysis was conducted with 100,000 simulated samples. Results indicated mean 30-day readmission costs: $16,037.08 (95% CI, $15,196.01 – $16,870.06) overall, $6,852.97 (95% CI, $6,684.44 – $7,021.08) for AMI, $9,817.42 (95% CI, $9,575.82 – $10,060.43) for HF, and $21,346.50 (95% CI, $20,818.14- $21,871.85) for THA/TKA” |
|
5 |
Why you choose Monte Carlo simulation? |
We appreciate the reviewers’ feedback for improving the manuscript. The revised manuscript incorporates substantial revisions to the methods and conclusion that seek to address the concerns expressed by Reviewer 1. This comment is addressed in lines 193-200 in the manuscript. “Our study sought to estimate the mean cost of 30-day readmissions, a probabilistic outcome subject to uncertainty due to various factors such as patient characteristics, treatment effectiveness, and healthcare processes. Monte Carlo simulation is well-suited for handling such uncertainty by repeatedly sampling from input parameter distributions to estimate the distribution of possible outcomes. In this case, the simulation would allow for the assessment of the uncertainty around the mean cost estimates and provide confidence intervals.” |
|
6 |
Table 1 is stretched over several pages, making it completely unreadable. I believe your task should be to present the results of the meta-analysis in a focused manner on key elements. Certainly, the construction of Table 1 needs reconsideration, and it might be proposed in a different form. Perhaps, some parts could be described. Figure 4 is unclear. |
We appreciate the feedback regarding the layout and structure of our article's charts and tables. We have considered your suggestions and made the necessary improvements. We have adjusted the layout structure of Figures 1,2 and 4 to align with the style of Figure 3, ensuring consistency throughout. Additionally, we have addressed the issue of Table 1 occupying excessive space by relocating it to the Annex. These revisions aim to enhance the clarity and readability of the visual elements in our article. Thank you for your valuable input, and we believe these changes will significantly improve the overall presentation of our research findings. |
|
7 |
This part is most important, you should describe more details: Finally, we simulated 100,000 sample draws from the modeled distribution for each related analysis for each readmission category simultaneously. We determined the weighted average of the included studies at each iteration. For each readmission category, we recorded the mean and 95 percent confidence interval obtained from the distribution of those 100,000 weighted averages. Monte Carlo simulations were carried out with the help of the Monte Carlo simulation software @RISK, version 7.6.1. (Palisade Corp). Following the Monte-Carlo interactions, the mean cost of 30-days readmission for all conditions is simulated at $16,037.08 (95% CI, $15,196.01 – $16,870.06). The mean cost of 30-days readmission for AMI is $6,852.97 (95% CI, $6,684.44 – $7,021.08). The mean cost of 30-days readmission for HF is estimated at $9,817.42 (95% CI, $9,575.82 – $10,060.43). The mean cost of 30-days readmission for THA/TKA is simulated at $21,346.50 (95% CI, $20,818.14- $21,871.85) |
Thank you for your guidance. The section you highlighted has been revised to provide more detailed descriptions of the methodology used in the meta-analysis, including the simulation process, determination of weighted averages, and calculation of mean costs with confidence intervals for each readmission category ( Pg 9, lines 267-281). “Finally, we conducted Monte Carlo simulations using @RISK software to analyze the data. Specifically, we simultaneously simulated 100,000 sample draws from the modeled distribution for each related analysis across all readmission categories. At each iteration, we determined the weighted average of the included studies. We recorded the mean and 95 percent confidence interval obtained from the distribution of those 100,000 weighted averages for each readmission category. Subsequently, we calculated the mean cost of 30-day readmission for all conditions and specific conditions, such as AMI, HF, and THA/TKA, along with their respective confidence intervals. Monte Carlo simulations were carried out with the help of the Monte Carlo simulation software @RISK, version 7.6.1. (Palisade Corp). Following the Monte-Carlo interactions as seen in figure 4, the mean cost of 30-days readmission for all conditions is simulated at $16,037.08 (95% CI, $15,196.01 – $16,870.06). The mean cost of 30-day readmission for AMI is $6,852.97 (95% CI, $6,684.44 – $7,021.08). The mean cost of 30-day readmission for HF is estimated at $9,817.42 (95% CI, $9,575.82 – $10,060.43). The mean cost of 30-days readmission for THA/TKA is simulated at $21,346.50 (95% CI, $20,818.14- $21,871.85).” |
|
8 |
The conclusions do not reference the results, are too general, and rather brief. They lack practical implications of the study. |
This comment is addressed in lines 367-390 in the manuscript. “Our study offers valuable insights into the financial burden imposed by re-admissions on acute care hospitals, providing robust estimates of attributable cost resources for readmissions across various medical conditions. By quantifying the mean costs of 30-day readmissions for conditions such as AMI, HF, and THA/TKA, our findings shed light on the economic impact of readmissions and highlight areas where healthcare resources are being allocated. One of the primary benefits of our study is its provision of concrete estimates that can inform decision-making and resource allocation strategies for healthcare stakeholders, including hospital administrators, policymakers, and payers. By understanding the financial implications of readmissions, stakeholders can develop targeted interventions and quality improvement initiatives to reduce readmission rates and optimize healthcare spending. Furthermore, our study opens avenues for future research by identifying persistent trends in readmission costs and emphasizing the need for continued efforts to address this challenge. Future studies can build upon our findings by investigating the effectiveness of specific interventions and strategies to reduce readmissions and improve overall healthcare outcomes. Additionally, exploring the impact of demographic and clinical factors on readmission costs could provide further insights into the drivers of healthcare expenditure. In light of the ongoing emphasis on value-based care and healthcare cost containment, our study underscores the importance of addressing readmissions as a critical component of healthcare quality improvement efforts. By addressing the financial implications of readmissions, hospitals can better align their resources and interventions to improve patient outcomes while optimizing healthcare spending.” |

Round 2
Reviewer 1 Report
Comments and Suggestions for Authors
The requested changes have been made.
Reviewer 2 Report
Comments and Suggestions for Authors
In the new version of the manuscript, the authors have made certain revisions, which I believe meet the requirements I have outlined. There are io further comments.